# Static and Dynamic Verification of Space Systems Using Asynchronous Observer Agents

**DOI:** 10.3390/s21134541

**Published:** 2021-07-02

**Authors:** Wiktor B. Daszczuk

**Affiliations:** Institute of Computer Science, Warsaw University of Technology, Nowowiejska Str. 15/19, 00-665 Warsaw, Poland; wbd@ii.pw.edu.pl; Tel.: +48-22-234-78-12

**Keywords:** modeling of space systems, verification of space systems, monitoring agents, observer agents, automated test generation

## Abstract

Formal verification of distributed systems is essential, especially in mission-critical systems that cannot be restarted. Such are space systems in which satellites read sensor values and autonomously make actuator decisions based on them, and ground services only set general patterns of behavior. The verification formalism should correspond to the essential characteristics of a distributed system, such as node autonomy and asynchrony of actions and communication, as in our Integrated Model of Distributed Systems (IMDS). It is also crucial that the formalism allows for finding partial deadlocks and checking partial termination, where only a subset of the system nodes is involved while the rest can perform their own tasks at the same time. This article presents the idea of using monitoring agents—observers prepared in the IMDS formalism. Observers check the state of individual system components by polling, allowing verification without knowing the global state of the system. Such an agent is an ideal prototype of a runtime observer that checks if the actual operation of the system corresponds to a design that has previously been proven correct.

## 1. Introduction: Model Checking of Space Systems

Satellite control is performed by distributed systems consisting of ground services, and the satellites work singly in groups. The satellites and planetary probes are built themselves as distributed IoT systems of cooperating nodes: sensors, actuators, controllers, and communication facilities. The system components work mostly autonomously, following defined procedures or scenarios, exchanging messages to define the goals and to monitor the operation. Such procedures need to be formally verified to ensure proper behavior. A poor example of a system which was tested but not formally verified was the Pathfinder Mars explorer that stopped its operation due to software error. Fortunately, the lander was modular and the components operated quite autonomously, which allowed it to communicate with the lander and load new software while it was physically immobilized [1]. The whole situation could have been avoided if the system had been formally verified before departure. Dijkstra has already stated that testing is for finding bugs, not for validating software [2]. First-year students know this well, but unfortunately, mature computer engineers often forget this basic principle.

We do not know how general the Pathfinder case is (but the priority inversion phenomenon occurs in many projects). Designers rarely boast about mistakes, but rather keep silent about them. However, even if Pathfinder were the only space mission that was threatened with a failure as a result of a process synchronization error, the possibility of losing several billion dollars and several years of work as a result of a programming error is a sufficient prerequisite for creating a methodology for formal system verification in the design phase, and later runtime verification of compliance with specification.

Model Checking (MC) is a verification of whether the behavior of a system fulfills the specification. The model can be a source code itself, with some reductions that allow for the elaboration of reachability space of behavior, or a transition system of a form of a graph. The specification is a set of properties given as algebraic formulas, or another graph. The verification is based on temporal formula evaluation or similar techniques [3,4]. In general, model checking requires:model of the state of the examined system; usually it is the valuation of variables of a specific program/set of programs. The natural requirement is the finiteness of the set of states.determining the sequences of states, that is, a succession graph. By transitively generalizing the succession, we obtain the reachability of states. Exhaustive search methods require this graph to be finite, but non-exhaustive methods do not necessarily.a language for defining properties. In the simplest MC this is defining the state that is desired/undesired, in a true MC this is finding a sequence satisfying certain conditions.

The contribution of this paper is our approach to verify distributed IoT systems, which consists of four principles:application of true asynchronous formalism because no synchrony is possible in physically distributed systems. The model should reflect natural features of a distributed system as the locality of knowledge and autonomy of decisions; consequently, the verification algorithms must support weak fairness (justice) and strong fairness (compassion) [5], which are essentially referring to asynchrony and autonomy;automated verification of basic features required in mission-critical distributed systems;verification concerning the situations in which a subset of cooperating nodes participate as opposed to total features (like total deadlock) specific for centralized systems;using monitoring agents whose purpose is twofold: (i) operation just like other system processes, without any non-local knowledge of states of the components; (ii) based on these agents, building testing processes that verify the desired features in runtime (because some elements of the actual system can diverge from the modeled behavior).

All the abovementioned elements of our approach are discussed in detail in Section 2.

The design of space systems includes numerous procedures and scenarios which should be formally verified to avoid the Pathfinder experience. Such a system should be verified in the design stage and should also be self-verifying in operation. In design, the entire global state of the system is known and can be subject to verification. However, it is evident that in run-time, no such global state exists, and no element of the system has knowledge about the operation of all components. It becomes clear that the global state assumed in the design state was only an illusion. Therefore, we introduce monitoring agents—called observers—that are defined and operate in the system just like other operational processes. We base the verification on the knowledge collected by these processes rather than on the global system state. Therefore, the correctness of the system decided on the observers’ behavior in the design stage is the same as the correctness tested dynamically.

Our goal is to build an asynchronous formalism and verification methodology for model checking of space equipment. The space equipment uses the readings on its own equipment and other satellites in Earth’s orbit (or elsewhere) with which it cooperates to set the values of the actuators. The cooperation of ground services, possibly distributed over Earth’s surface, is needed. The typical approach used in model checking is based on the knowledge of the entire state space of the verified systems, even if its components are aware only of their local state and received messages. This approach is successful and gives knowledge on possible deadlocks and other malfunctions. However, when monitoring the real system in operation, different methods must be used because the global state cannot be acquired. Instead, we use the observers that interrogate the devices for their states and actions and conclude about correct or erroneous work. Those agents are used both in verification and in operation. Therefore, the same model is used to prove the proper system design and investigate if actual operation follows the design rules used in the project.

For the model checker, the entire state space (which is called the reachability space in our case, for reasons explained in Section 2) is known, because in the verification some global properties are identified even if they are observed from the point of view of individual components in run-time. We postulate adding special monitoring processes called observers that have only limited knowledge about the system state. An observer collects this knowledge by interrogating system modules about their progress. Thus, the correctness of the system can be judged based on the observers’ behavior rather than by decisions based on the system global state (which is nonexistent in distributed system operation). Having such observers defined, we can include them into a running system after their transformation (manual or using a software tool) to acting processes. Their purpose is to check if the system components follow the specified behavior. If yes, observers confirm the correct system run at the time. If not, we have an early warning that the system is expected to diverge from its proper operation, or even that it already works incorrectly.

This is comparable to automated test generation from the specification, which is well known from the literature (references are given in Section 8). However, the role of the tests performed by the observers is different: checking if the system components follow the behavior that is statically proven to be correct rather than validating whether the system states follow the abstract specification. This rule, together with using a formalism specially tailored for distributed systems, automated verification, and checking partial features, demonstrates the novelty of the approach.

We verified numerous distributed systems, as other research teams have. Many formalism and verification tools use the synchronous or quasi-synchronous paradigm, even if they are called synchronous. They often lack other properties inherent to distributed systems, like autonomy, locality, and checking partial features. The tender for a verification environment for space equipment AO10416, announced by ESA (European Space Agency), inspired us to present our approach in the context of systems that are extremely distributed. This problem highlights the presented characteristics of our approach. The analysis of the nature of the space system led us to the invention of observers: the migrating agents whose progress is the base of verification both in static form and in run-time.

The sequence diagram of the system to be verified is presented in Figure 1. It copes with a satellite operating autonomously when started by ground services (called the “environment”). The satellite has an instrument bay whose door is opened when it goes operational. The door is closed if the sensors report a dangerous situation. In such a case, the door should be closed. The tender states that the system cannot be in operational mode when an alarm has occurred. The door should close if “more than two alarms” are recorded. The requirement is inconsistent with the sequence diagram, in which precisely two alarm signals from sensors cause the closing of the door, rather than more than two. The original graph can be found at http://www2.rosa.ro/index.php/ro/esa/oferte-furnizori/4628-model-checking-for-formal-verification-of-space-systems-expro (accessed on 30 June 2021). The abbreviation FDIR is not elaborated anywhere in the ESA tender materials, so we treat it as a proper name.

The idea of static verification and dynamic testing on the same model with monitoring agents is important when making updates to the software, if changes are made in ground services software, or uploaded to satellites’ software. This is typical when new experiments are planned for existing equipment or its mode of operation is modified, or if the sensor readings cause a different action of actuators. Using the idea of observers, the new way of work can be checked statically for errors and then applied to acting devices to test if they follow the assumed changed behavior.

## 2. Informal View on Integrated Model of Distributed Systems

Our modeling formalism is specially tailored for distributed systems, as it reflects their natural features: locality of actions, the autonomy of decisions, and asynchrony in several aspects. The distributed systems used in space equipment operate locally, autonomously controlling their devices by employing sensors and actuators on general demand acquired from ground services. They report their current state and actions to ground services in the reverse direction. The mode of communication is of asynchronous nature, i.e., the components of the system operate locally and send asynchronous, non-blocking messages to each other, hoping they get an answer. The answer informs the caller about the reporter’s state at the moment of issuing, which can no longer be valid when the message reaches its destination. Weak and strong fairness in verification, which reflect the asynchrony and autonomy, liberate from false deadlocks resulting from postponing the progress of some system components.

Synchronous communication based on pairs of asynchronous messages can be built on a higher level of abstraction and upon lower asynchronous communication. However, such communication cannot be considered in space systems. For example, the entire round time of communication with Mars explorers takes more than 20 min.

Automated verification concerns a subset of features of distributed systems. At the cost of limiting the verified properties, the checking in “push the button” style is available, which does not require the designer to learn of formalisms like temporal logic. We decided to choose four properties: communication deadlock, resource deadlock, the inevitability of distributed termination, and the possibility of distributed termination. Deadlock is a feature automatically searched in many model checkers, for example, Spin [6] or Uppaal [7] (however, they do not distinguish between deadlocks in communication and deadlocks over resources). Other properties like distributed termination are seldom automatically verified, and if so, the shape of the system model is limited, for instance, to cycling systems. We have included the possibility and inevitability of termination in systems of arbitrary schemes to automatically verified features. For verification, the Integrated Model of Distributed Systems has been elaborated, in which a distributed system can be specified and two views of the system—communicating nodes or traveling agents—can be observed and verified.

The most important aspect of verification in IMDS is the verification of situations in which a subset of system processes participate. These properties are partial deadlock and partial termination. They are crucial in verifying distributed systems, as several processes can fall into a deadlock while the rest of the system operates. According to termination, it is expected that not all components of the system are supposed to terminate, as the system can contain components performing some services continuously, and other components which finish their operation on completion of their mission. We believe that the enumerated set of properties—including partial features—is enough, because typically other kinds of behavioral errors finally manifest themselves as deadlocks or lack of termination. Of course, this type of verification is not suitable for most data errors, for which different kinds of checking are addressed.

There are many model checking methods, and the most advanced are those based on temporal logic, as they allow properties to be expressed in different states or sequences. Model checking is performed over the reachability graph of the system, which consists of the vertices being system states and transitions which are actions. Every model checker uses a single graph for verification, on which temporal formulas are evaluated, or a similar checking mechanism is applied. Of course, in a distributed system such a graph must be elaborated as a net effect of a composition of individual components’ graphs. The global graph is stored in the checker or calculated partly using non-exhaustive verification methods. However, it is never directly observed as a whole due to its size (except for tiny cases).

A verification method based on the IMDS (Integrated Model of Distributed Systems) formalism has been developed in the Institute of Computer Science of the Warsaw University of Technology [8]. It addresses primarily the verification of distributed systems. Both the state of distributed components (called in formalism as servers or modules) and messages on the way or received and waiting for acceptance must be taken into account. This is obvious because the behavior of the module depends not only on its current state, but also on what messages are “on the input.” Different messages can trigger different module actions. Therefore, a node in the reachability graph is described by the state values of all modules of the distributed system and all messages in transit or waiting at the module input. To distinguish such a graph vertex from a state, which is a combination of local states of the modules, we call it a configuration. We call a global graph a reachability space, rather than a state space, for the same reason.

The idea of IMDS is based on defining:set of module states; a module can be in one of its local states,a set of messages that can appear at the input of the modules; multiple messages can be pending at the input of a module, and they are not structured in any way: they are simply the input set of messages,distributed module is considered as a set of actions; action is a call to a specific module service by a message on its input: executing an action causes the module’s state to change to the new one, then the message is consumed and a new one is generated on the action output. This message can be directed to the same or (usually) to another module.

In the theory of distributed systems, two basic paradigms of computing are considered:client-server model, in which we distinguish modules offering services (servers) and modules that trigger these services using messages (clients). The server may be a client of another server;a remote procedure call (RPC) model in which processes move from one module to another to execute a specific procedure in a target module, and the target module may call for further procedures on subsequent modules.

The IMDS formalism combines both models by introducing the concept of *agents*; these are comparable to the processes considered in the RPC paradigm. A module is represented by its set of states, and the agent is represented by its set of messages. It is assumed that the module action changes the state to the new one, and at the same time, it changes the input message to the output message belonging to the same agent. This rule ensures the continuation of both the module and the agent. A distributed system is simply a set of module actions in the context of sets of states and messages. The set of actions can be decomposed into subsets. Suppose it is decomposed into subsets of actions of individual modules: it is a *module view*, in which modules cooperate using messages (as in the client-server paradigm), while if it is decomposed into subsets of actions of individual agents, we get an *agent view* similar to the RPC paradigm (however, this is more general than RPC, as the agent does not need to return to the invoking module). Thus, in the formalism, every node in the node view is identified with a node process consisting of actions having this node state on input. Likewise, the agent in the agent view is a set of all actions invoked by messages with this agent identifier. Agent messages are the carriers of agents, while node states are the carriers of node processes. The details of IMDS are given in [8]. Regardless of the adopted view, it is still the same system defined as a set of actions. It is an implementation of the famous Lauer-Needham postulate of two equivalent implementations of a parallel system based on data (states) and messages [9]. However, the IMDS formalism goes even further, proposing one uniform system with two views.

In IMDS, messages “on the way” are not modeled—the sent message is immediately placed on the input of the target module. However, this does not apply to the real-time verification in which the message flow is modelled, which is discussed below.

The IMDS formalism adopts the interleaving semantics of the system behavior, which means that only one action is performed at a time. In a distributed environment, this may seem to be too strong a limitation, but the coincidence semantics can be transformed into an interleaving one [10].

A significant feature of IMDS is its asynchrony: note that in an action, the message is sent without caring about the current state of the message’s target module, which reflects the actual way of cooperation between distributed Earth and space nodes. It is unimaginable that the sending module guesses the state of the target module. Of course, the module can ask another module about its status using a message, but after receiving the response, it is not even known if the other module is still working. Many verification systems are based on synchronous formalisms, but they are unrealistic for distributed cases. In synchronous models, components agree on their states to communicate. The examples are Büchi automata [6] and Timed Automata [11], in which the components perform their local actions independently, in arbitrary order, but communication (implemented as joint actions on channel variables) requires synchronized common transition of both components. In those models, synchronization is used to communicate, while in our approach, communication is used to synchronize.

Model checking in IMDS is performed as an evaluation of temporal logic formulas over the reachability space. In particular, temporal formulas for a deadlock have been defined [12]. In the module view, a deadlock is such a state of the server that a different state will never replace, i.e., no action will be performed. An additional condition is the presence of a message at the module’s input because the lack of any action in the future means that the message will never be handled. The lack of messages in the module (now and in the entire future) means that the module is simply idle.

In the agent view, a deadlock means that the agent has a current message on the input of a module, but that module will never perform an action for that message (it can perform actions for other agents). Typically, an agent deadlock is also a module deadlock, but not necessarily: the agent may be deadlocked due to lack of resources for it in the system, while other agents are working (their actions are being performed), and no module is deadlocked.

This definition of deadlocks is fundamental because a deadlock does not have to be total, as in most formalisms [13]. It is common for some modules/agents to deadlock in a distributed system while others are still running. We call such a deadlock partial. Total deadlock is just a partial deadlock of all processes (modules or agents).

There are only three features verified in IMDS: distributed (partial) module/agent deadlock, the inevitability of distributed (partial) agent termination, and the possibility of distributed (partial) agent termination [13].

Partial deadlocks are challenging to identify. Usually, the methods of verification described in the literature find them only for systems with a specific general structure or leave the appropriate temporal formula to be written by the user. Formulas developed for IMDS are general and structure-independent and can therefore be evaluated automatically. The limitations of temporal verification to deadlocks (and termination) may seem like a significant limitation to the user, however:deadlock is a frequent design error, especially in distributed systems, and it is a fatal error (out of approx. 300 students who checked their solutions for synchronization tasks, approximately 10% detected deadlocks, even though the teachers previously positively assessed these solutions);other types of errors, such as performing operations in the wrong order, leads to later deadlocks;if we check the system for reaching a particular situation or completing a specific sequence, then at the end we can put a “deliberate” deadlock or put a process termination that the verifier will check; we often use both techniques.

An essential result of the temporal verifier operation, apart from the evaluation of the temporal formula, is a counterexample which is the history of the run of the verified system from the initial configuration until the error occurred. In this way, the designer is assisted in finding bugs and can quickly fix the system. In the case of a distributed system, the counterexample is a sequence of state changes and messages transmitted that can best be represented in the form of a message sequence chart.

The IMDS formalism has a timed version (T-IMDS [14]) in which time ranges can be attributed to actions executed in nodes, for example, the time between the signal from a sensor to the reaction of the actuator, and to messages sent between the nodes. This allows verifying if the system acting in given time conditions works appropriately. It is crucial because time constraints can prohibit some deadlocks, but new deadlocks can arise.

We have successfully verified many distributed systems, including aviation equipment. We have identified numerous errors and ambiguities in those systems. Until now, we have no experience with space equipment, but we are sure that our IMDS formalism, automated verification, and the idea of observers match the requirements for space systems verification.

## 3. Monitoring Agents: Observers

In an actual space system, it is impossible to obtain a global state. We can observe such a state (precisely: a configuration of IMDS) in the model, but other methods not based on the global state must be applied for online testing. No global state exists in a disturbed system. To match the model for static verification with the online testing model, in IMDS we can introduce additional monitoring agents called observers. The verification can be based on the behavior of those agents, which have only limited knowledge about the system operation. Namely, they have only the knowledge collected by interrogating individual modules about the values of sensors and the actions performed by the actuators. The observers act as ordinary agents and use a regular format of messages to migrate between modules. The operation of the modules is disturbed by observers only slightly, as they answer the observers’ questions about the actions performed.

If we prove the properties of the system based on the behavior of observers, we can implement the actual observers in the running system. The run-time observers act as ordinary agents just like in the verification model. If an action is skipped or executed in the wrong order, the observer can report and prevent damages that follow improper behavior. We can say that a model is a prototype of the actual space system, while observers are prototypes of online testing equipment.

Let us consider the idea of an observer collecting some events, presented in Figure 2. Suppose that there are two modules *S1*, *S2* representing the satellites and a module *G* representing the ground services. The usual cooperation between the modules is shown as bold arrows because the exchange of messages between these modules is not important from our point of view. It is important that these modules reach the correct states in the correct order. This is the case in the ESA tender, where the way of cooperation between the *MM*, *FDIR,* and *IM* modules is not defined at all. Only alarm signals from sensors that are counted are included. The observer *O* interrogates the module *S1* for some event *E1*, and if it gets a notification, it asks the module *S2* if it undertakes an action *E2* that should follow *E1*. The sequence of messages that together make up an observer agent is shown as thin arrows with sequence numbers of proper behavior.

The static verification consists of confirming:If any deadlock occurs in the system.If the observer terminates successfully after reporting the appropriate sequence of events.

## 4. The Model Checking Environment

The Dedan [15] environment was built in the Institute of Computer Science of the Warsaw University of Technology for verification in the formalism of the IMDS. The Dedan environment includes:the subsystem for constructing the reachability space,temporal verifier,counterexample editor,compiler of the imperative language Rybu to the IMDS notation [16],compiler of the BPMN workflow specification language to the IMDS notation [17],graphic editor for the specification of modules and agents in graphic form,reachability space viewer,simulator operating on configurations,graphic simulator operating on graphic images of modules (does not require elaboration of the reachability space),model export subsystem to Spin, NuSMV, and Uppaal model checkers.

The figures that are the output of verification are shown in sections on verification of an example: counterexample sequence diagram in module view and agent view, counterexample simulation over reachability space or space examination, simulation over counterexample configuration trace, simulation of counterexample over states and messages or arbitrary simulation, and finally graphic counterexample simulation over DA^3^ automata or arbitrary simulation over automata.

In addition to academic examples, the Dedan environment was used to verify medium and large systems, such as monitoring protocol in the SCADA (Supervisory Control And Data Acquisition) system (a deadlock was detected in the protocol), three-stage production pipeline, communication protocols in a system for plant identification and control of an industrial facility, streaming protocol for plant monitoring using SMS (Short Message Service) messages of the GSM (Global System for Mobile communications) network, traffic light controller, and autonomous vehicles cooperation.

The following verification of large systems was performed:timed verification of the Karlsruhe Production Cell [18]; it has been shown that the proper selection of time intervals for the operation of individual cell devices allows for its safe operation; improper selection of intervals causes problems (e.g., failure to collect the part from the conveyor on time causes the part to fall onto the floor);timed verification of the Rome metro model [19]; in a timeless model, a critical region is necessary that limits the number of trains in a given area; in the timed model, it has been shown that it is enough to properly select the intervals of passing sections of the track and stopping time at stations.

Both the construction and search of the reachability space is a very time-consuming task. Therefore, work is currently underway on the distribution of calculations with non-exhaustive verification using the HTCondor tool and the Shapp library [20].

## 5. Model Checking an Example from ESA Materials—Naive Model

We used DA^3^ Distributed Automata notation for modeling, which is semantically equivalent to IMDS [21]. The automata are of Mealy style, i.e., the output symbols are generated on transitions. The rules of the automata construction are:every node is implemented by an automatonautomaton vertices are node statesinput symbol triggering the transitions are incoming messagesoutput symbols generated on transitions are outgoing messagesinitial vertex of an automaton, pointed by dot-rooted arrow ●→, is the initial state of the nodeevery automaton is equipped with a set of pending messages (shown as a penny cress in the figures)initially, the message set contains initial messages of all agents started in the nodemessages have the form *ag.nod.serv*, which means that the message calls the service *serv* on the node *nod* in the context of the agent *ag*. Input and output messages are separated by a slash. Minus character means agent termination—no continuation message is issued.

The set of figures contains the automata used in the example implementation. Our model consists of the following modules:Environment, representing ground services, the *SA* (Starting Agent) agent is initiated in it, which causes the entire system to run,*MM* (Mode Manager), managing the state of the satellite, where *IMA* (Instrument Manager Agent) agent is launched—managing door closing and *CR* (Counter Resetter)—resetting the alarm counter,*C* (Counter)—alarm counter module, it is implemented separately from the MM module, because the combination of *MM* and *C* in one module would result in a synchronous change of variables responsible for the device state and the alarm counter; in that case, the verification would be useless,*FDIR*—control module generating alarms (at random times); the *AR* agent (Alarm Reporter) is launched in it; this agent represents signals from the sensors, for example, meteor shower sensor, overheating sensor, etc.,*IM* (Instrument Manager)—door closing module,Observers—a module that checks if the correctness condition is met by asking *MM* and *C* modules for their states; the *MMSR* agent monitoring *MM* and *CSR* agent monitoring *C* are activated; since the Environment module is responsible only for starting the system, the Environment and Observers modules have been combined into one *EO* module.

The first implementation is “naive”, which is discussed below.

The *MM* module (Figure 3) starts idle, the agent *SA* message received from the *EO* module takes it to the *starts* state, from where the agent *CR* puts it into the *oper* state after sending the *reset* message to module *C*. After receiving the *alarm* signal from the sensor, the *AR* agent sends a counter-increment message to module *C* and transfers *MM* to *fail* state. The *AR* agent, after receiving confirmation of the counter-increment by the AR agent, sends the appropriate acknowledgment to the *FDIR* from which the alarms came. The next alarm from a sensor preserves the *MM* module *fail* state and causes a consecutive counter-increment sequence. Confirmation from the counter causes the transition to the *shut* state in which the door is closed by the actuator—the *IMA* agent sends a *close* signal to the *IM* module and causes the *MM* to go to the final state *finish*. Another alarm does not change anything, so the reporting agent is terminated.

In each of the states, the monitoring signal received by the *MMSR* agent from the *EO* module causes the *MM* module operating phase report to be returned to the *EO*: *befop* before the *oper* state, *op* in the *oper* state, and *nop* after the *oper* state.

Module *C* (Figure 4) is initialized to *undef*, after which the agent *CR reset* signal takes it to state *er0* (no errors). Each *inc* signal of the *AR* agent increases the error counter by 1 (to *er1* or *er2*), and the next alarm from the sensor (the third one) leaves the module in the state *er2* and terminates the *AR* agent.

In each of the states, the monitoring signal received by the *CSR* agent from the *EO* module causes the *C* module operating phase report to be sent to *EO*: *ne1* in state *er0*, *e1* in states *er1*, *er2*.

The FDIR module (Figure 5) reports the alarm signals from sensors. The module is initially in a state *na* for an indefinite amount of time (self-transition), after which it notes an alarm at a random moment by sending an *AR* agent message to the *MM* module which goes to state *a*. After receiving an acknowledgment, the module returns to the state *na*.

The module *IM* (Figure 6) receives the *close* signal from the *MM* module and terminates the *IMA* agent. In fact, this module starts closing the door by issuing a signal to the actuator.

The *EO* module (Figure 7) is initialized in the *ini* state and the *SA* agent *start* signal starts the *MM* module operation and causes the transition to the *wait* state. The module enters the *ent_oper* state upon receipt of the *op* signal sent by the *MMSR* observer. Each subsequent *op* signal causes it to stay in this state as well as signal *ne1* from the counter monitor. Signal *nop* means correct *MM* exit from *oper* state, and further events do not matter as the module reaches *left_oper* state. The signal *e1* means that the counter *C* goes to the state *er1* (one alarm), which means failure to meet the condition: there are two alarms, and the *MM* module did not exit the *oper* state. The *EO* module goes to the *fatal* state, and the *CSR* agent sends an *error* signal to the same module. Since the *error* signal does not cause any action, this is an “artificial” *EO* module deadlock signaling an error.

There is an inconsistency in the ESA materials regarding the number of alarms: in the informal description, the door is closed after >2 alarms (i.e., after 3), and in the message sequence chart after 2. However, this does not affect the verification because we are dealing with the first alarm (stop condition for alarm counter >0).

Some additional actions in the module not shown in the sequence diagram handle some invalid sequences, for example, receiving the *befop* after the *op* signal. They all lead to a *fatal* state. They are useless in the verification of proposed requirement but can be useful in on-line testing of the equipment.

## 6. The Verification

This model appears to be correct. However, the verification shows the possibility of an error. Figure 8 is a message sequence diagram of the counterexample. In this view, every node has a timeline beginning with its initial state, and below the starting messages of all agents in a given node are shown on a yellow background. Then, every action is presented as the new state of the node in blue, the agent executing the action on light blue, and the message by which the agent continues its run using an arrow and the message target on message destination node on yellow. Thus, the migration of the agents back and forth between the nodes is highlighted. At the bottom, states of all nodes and final messages of all unterminated agents are presented. The deadlock denotes that there are messages pending on the *EO* node which will never be served. It is an “artificial” deadlock, introduced to distinguish the erroneous situation from the correct one, in which the observers simply terminate.

The same error in the agent view is seen in Figure 9. In this view, the node timelines are shown on the left with ping headings and the agent timelines with green headings on the right. Below, initial states on the nodes and initial messages of the agents are included. Then every action has a form of specification of the current node of the agent in blue on its timeline, and the new state of the node on the same level in light blue (with the name of the agent causing this new value of the state), and the arrow of the message itself with target node in light yellow and the message name in yellow. Like in the node view, the current states of all nodes and final messages of all unterminated agents are given at the bottom.

Several forms of the reachability space viewing and simulation are elaborated: Figure 10 shows the entire reachability graph. Watching this graph is reasonable for small graphs only. On the right, it shows all configurations and actions as transitions between them. A configuration consists of states of all nodes and messages of all agents. To distinguish them, they are arbitrarily numbered. During the simulation over the reachability space, the “current” configuration becomes green and the outgoing transitions are red. This “current” configuration is also shown on the right, and below are transitions numbered arbitrarily. Every transition (in pink) shows a state and a message that triggers the action. In light blue, the changes in configuration are shown: a new state of the node and a new message of the agent. Clicking a transition causes this target configuration to become “current.” The simulation can follow a counterexample presented in Figure 8 and Figure 9, or the user can diverge from the counterexample in an arbitrary moment.

Figure 11 presents the simulation over configurations of the system, which is more frequently used because it does not require generating the entire reachability space in the graphic window. In fact, it does not require calculating this space at all. Thus, this mode of simulation can be used even if a non-exhaustive verification method is applied. In the window, the states of all nodes in the current configuration are shown on the top and agent messages below them in pink. The pairs of (*message, state*) triggering the actions are shown as pairs of arrows ending in the target configurations in light yellow, where a new state and a new message are included. This mode of simulation can follow a counterexample or diverge from it, and backtracking is also possible. Figure 12 shows an intermediate simulation mode which has some features of both modes presented in Figure 10 and Figure 11.

Finally, Figure 13 shows the simulation over Distributed Automata that are formally equivalent to IMDS [14]. The actions change the current state of an automaton and switches the focus to the automaton being the target of the message generated in the action. The user can then follow the agent of switch until another, arbitrarily chosen action is selected for automaton. This mode of simulation is most often used and it does not require calculating the system reachability space. Just as previous simulation modes, this mode can follow the counterexample achieved from the verification.

All the presented screenshots concern the example system presented in Figure 3, Figure 4, Figure 5, Figure 6 and Figure 7. Coming back to this system, the error results from the fact that the *EO* module has not yet managed to ask the *MM* module for its state, while the information about two alarms has come from the sensors. This is due to the asynchronous mode of operation of the system. An assertion could check the correctness condition, but the Dedan environment does not provide such a possibility. On the other hand, assertions can only be used for validation in a laboratory environment where the internal states of all modules are accessible. If the *Observers* were to be an actual set of processes working in the system and detecting errors, they must act precisely by querying modules for their states. Therefore, we modified the basic model which allows the system to be examined by an observer operating in it, equivalent to test generation.

The Appendix A and some basic examples of IMDS specification can be found at the URLs provided below the article.

## 7. Verification of the Corrected Model

In the corrected model, it should be ensured that the information about the leaving of the *oper* state by the *MM* module reaches the *Observers* before the *MM* module sends a signal incrementing the counter. It is necessary to enter the intermediate states and a continuation signal *cont* in the *MM* module. This is shown as actions in the IMDS source notation. This code contains a record of actions grouped in individual modules, plus the necessary syntactic glue (module types, module, and agent variables, binding formal module parameters to actual ones, initializing modules and agents, etc.). Figure 14 contains the transcription of the DA^3^ transition to the action.

These new, intermediate states are used in the following actions:

MM:

{AR.MM.alarm, MM.oper} -> {AR.MM.cont, MM.fail_inf1},

{AR.MM.cont, MM.fail_inf2} -> {AR.MM.cont, MM.fail_inf2},

{AR.MM.cont, MM.fail_inf3} -> {AR.MM.cont, MM.fail_inf3},

{MMSR.MM.rep, MM.fail_inf1} -> {MMSR.EO.op, MM.fail_inf2},

{MMSR.MM.rep, MM.fail_inf2} -> {MMSR.EO.nop, MM.fail_inf3},

{MMSR.MM.rep, MM.fail_inf3} -> {MMSR.EO.nop, MM.fail},

The new model does not show deadlocks, which is presented in Figure 15A.

In the corrected model, in which no deadlock happens, it can be checked whether the second requirement holds, i.e., the door should be closed if sensors report two alarms. We assume the fairness of the model, which means that if two alarms can happen, they must happen (preceded possibly by many situations in which they do not happen). We check this property by simple verification for termination of *IMA* agent, which is expected after two alarms. This agent terminates after closing the door, which is specified as the desired result in the original ESA materials. The result of the verification is presented in Figure 15B, and the witness sequence diagram is shown in Figure 16 (node view) and Figure 17 (agent view). These graphs show that, according to the ESA tender requirement, the *IMA* agent successfully terminates after closing the instrument door (in green ovals) in response to two alarm signals from the sensors (in red ovals). In our opinion, an additional observer can be added to the system which successfully terminates in the *EO* module after interrogating *IM* for the successfully closed door.

The source codes of the observers are simply the tests for the system operation. On the example of CSR, which code extracted from the agent view is determined as follows:


agent:    CSR (servers EO:EO,C:C),
actions   {
    {CSR.EO.ne1, EO.wait}      -> {CSR.C.rep, EO.wait},
    {CSR.EO.e1, EO.wait}        -> {CSR.EO.error, EO.fatal},
    {CSR.EO.ne1, EO.ent_oper}        -> {CSR.C.rep, EO.ent_oper},
    {CSR.EO.e1, EO.ent_oper} -> {CSR.EO.error, EO.fatal},
    {CSR.EO.e1, EO.left_oper} -> {EO.left_oper},
    {CSR.EO.ne1, EO.left_oper}        -> {EO.left_oper},
    {CSR.EO.e1, EO.fatal}       -> {CSR.EO.error, EO.fatal},
    {CSR.EO.ne1, EO.fatal}      -> {CSR.EO.error, EO.fatal},





    {CSR.C.rep, C.er0}     -> {CSR.EO.ne1, C.er0},
    {CSR.C.rep, C.er1}     -> {CSR.EO.e1, C.er1},
    {CSR.C.rep, C.er2}     -> {CSR.EO.e1, C.er2},	
};

The agent automaton of *CSR* is presented in Figure 18. The vertices of the automaton are messages (sent from *EO* on the left and from *C* on the right), while the labels on transitions are the context in which the messages are sent (if the state does not change, for example, *EO.wait*) or state changes of the modules if it happens in the action (for example *EO.wait*->*check*). Additional dashed vertices in the bottom represent the termination of the agent (no message is issued) and its “artificial deadlock.” The automaton of the *CSR* or *MMSR* agent can be programmed in a language used to implement the system. In order to conform to the formal model, the transmitted messages must be provided with the identifier of the agent in the context in which they are sent. For example, see messages in the above *CSR* agent process which have this identifier as the first item.

Such a construction of the automaton may seem strange for people familiar with automata in which the vertices represent states and the transitions are events (like messages). However, this is simply a dual view of the system: in the node view, the vertices are states and the transitions are messages, while in the agent view, the vertices are messages and the transitions move agents from a module to another one, with a possible change of module state. For details, the reader can refer to [8].

The *CSR* agent migrates from *EO* to *C* and back. It interrogates the *C* node for the number of alarms reported by sensors. After receiving the report on two alarms, it checks the current state of *EO*. The state *left_oper* denotes a proper sequence, thus the agent simply terminates. If the state is fatal, an “artificial” deadlock is introduced: the agent sends the message *error* which is never served. This automaton can be translated to a program in the pseudocode below. Note that the state of *EO* is modified by this agent (*state*=*fatal*) and by other agents, for example MMSR sets *left_oper*. Successful termination is translated to *terminate*(0), while the artificial deadlock to *terminate*(-1).

In *EO*:


reply=ne1;
while (state in {wait,ent_oper} && (reply==ne1))
{
    send(CSR,C,rep);
    receive(CSR,C,reply);
}
if (reply==ne1)
    if (state=left_oper) terinate(0);
    else terminate(-1);
else terminate(-1);



In *C*:


state=er0;
while (1)
{
    receive(CSR,EO,reply);
    switch (state)
    {
    case er0: send(CSR,EO,ne1);
    case er1: send(CSR,EO,e1);
    case er0: send(CSR,EO,e1);
    }
}

The code of the agent can be stuffed with assertions that control correct succession of states of a node for *MM* (reported by *MMSR*): *idle* -> *starts* -> *oper* -> *fail* -> *shut* -> *finish*, messages of the agent, for *CSR*: *ne1* -> *e1* -> *term*. Some states/messages can be skipped, but there is no way back in these sequences. Any deviation from these rules denotes an error.

## 8. Related Work

There are asynchronous formalisms, but usually the asynchrony is understood in the wrong way. Authors call the communication asynchronous if it is non-blocking, and this definition is close to ours. It is often assumed that the state of the mating node must be known for communication to occur, regardless of whether it is blocking. In a really distributed system, such an assumption is unrealistic: how can we check the state of a remote note using another form of communication? The third notion of asynchrony relates to the actions executed in distributed nodes. Models like Büchi automata and Alur’s Timed Automata [22] execute common transitions synchronously, as common symbols trigger them. Those automata are called asynchronous because the other, unrelated actions executed locally do not interfere. However, in our opinion, such systems should be called synchronous. Even if any one of the nodes can wait for the other one to enter a proper state, how can we check it? Finally, a synchronous system can be defined as one in which all distributed components execute in lockstep using the same central clock. Therefore, an asynchronous system is one in which all distributed components execute out of lockstep following their own local clocks. In our approach, we do not introduce any notion of clocks, neither central nor local.

There are several formalisms for modeling distributed systems. Most of them have their roots in the specification of centralized systems, then are extended to cover distributed environments. Therefore, most of them are synchronous, or the synchrony is loosened using a kind of inter-process buffer [23]. A good example is the verification of Avionic systems in Spin [24]. For communication, Promela channels are used. This is not a problem when a single process uses a channel. However, if multiple processes communicate using the same channel, this approach artificially schedules messages, which may itself cause errors. To our best knowledge, our formalism applies asynchrony in the most consequent way. An example of a quasi-asynchronous model is the classical paper of Havelund et al. [25]: the authors use Promela, in which the messages are buffered in bounded FIFO channels. This paper is similar in using a remote agent that can be compared to our observer. The disadvantage of that paper is using Spin [26], in which only total deadlocks are automatically searched and strong fairness is not supported, sometimes leading to false deadlocks [5]. Strong fairness should be supported in model checking of distributed systems, because assuming some influence of a module on the process scheduling in other modules is unrealistic. We have not found any verification environment based on full asynchrony and strong fairness which is natural to distributed systems.

Automated test generation based on verification is presented in symbolic execution methods [27,28]. Tests generated from various abstract models are described in [29,30,31,32,33]. Tests generated from timed automata specification (which are synchronous specifications from our point of view, since they are based on executing common actions in parallel components) are covered in [34]. Tests generated from ontology specifications are presented in [35]. Automated test generation from layered models is described in [36]. However, none of the approaches use the same processes addressed for static specification without global knowledge of the distributed system state and the same processes acting as agents in the run of the actual system.

Please note the different purpose of our observers compared to “traditional” tests. Usually, the tests check if the system or its components reach desired results. In our approach, the system is proven statically correct if the event succession follows the specification and the tests check if there is no divergence from the assumed succession of events.

## 9. Conclusions and Further Work

We showed a verification of a space system in which signals from sensors initiate appropriate reactions. The model highlights the actual features of communication with satellites and between satellite modules.

The novelty of our approach lies in the connection of several techniques which reflect the actual properties of this structure and operation of highly distributed space systems:true asynchronous formalism, which highlights issuing messages without any knowledge of the state and operation of the target node;expressing locality of decisions and autonomy of actions in distributed components;automated verification which supports checking the correctness of many versions of the system, without any knowledge on temporal logics;strong fairness of the evaluation algorithm, protecting against false deadlock resulting from assumed scheduling of processes;checking partial features—deadlock and termination—in which a subset of system processes participate;original concept of observer agents which can verify the correctness of the system statically and dynamically in run-time.

We showed how the example system published by ESA can be verified. However, this verification is timeless, and the next step in our research is real-time-based verification, considering possible boundaries of message delivery and action duration. Such a real-time related verification in other models is presented in [31,34].

## Figures and Tables

**Figure 1 sensors-21-04541-f001:**
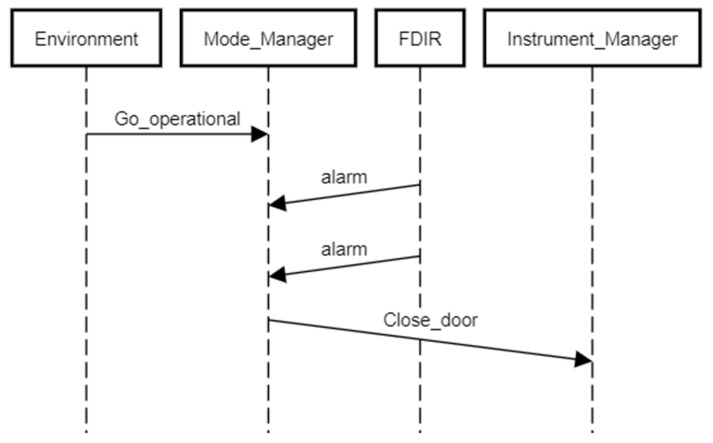
The sequence diagram of the example space control system.

**Figure 2 sensors-21-04541-f002:**
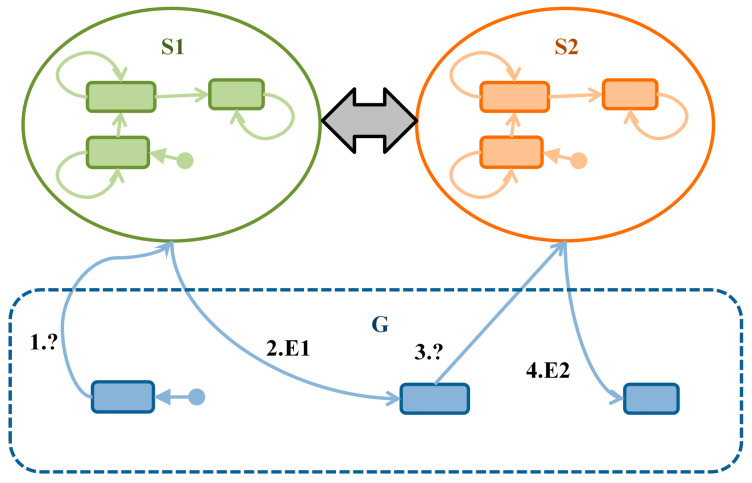
The idea of an observer interrogating two modules *S1* and *S2* for their states.

**Figure 3 sensors-21-04541-f003:**
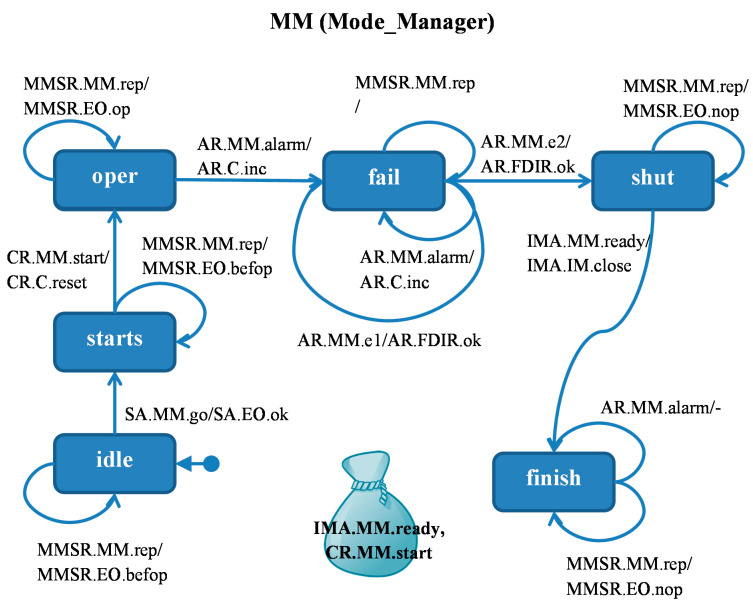
*MM* module. Agents: *IMA* (Instrument Manager Agent), *CR* (Counter Resetter), *AR* (Alarm Reporter)—starts in *FDIR*, *SA* (Starting Agent) starts in *EO*, *MMSR* (MM State Reporter)—starts in *EO*.

**Figure 4 sensors-21-04541-f004:**
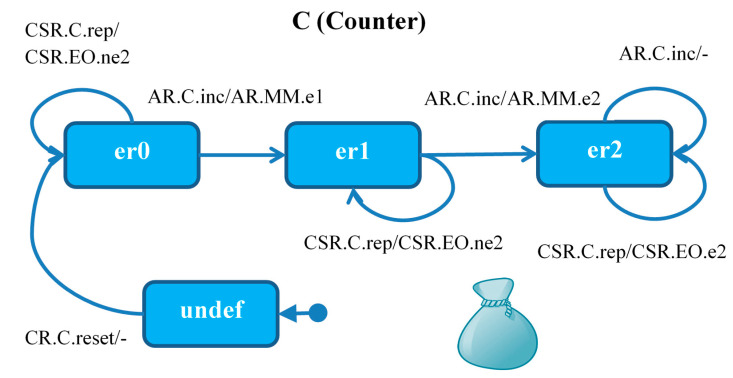
Module *C*: agent *AR* (Alarm Reporter) starts in FDIR, comes through *MM*. Agent *CR* (Counter Resetter) starts in *MM*. Agent *CSR* (Counter State Reporter) starts in *MM*. Initially the message set is empty.

**Figure 5 sensors-21-04541-f005:**
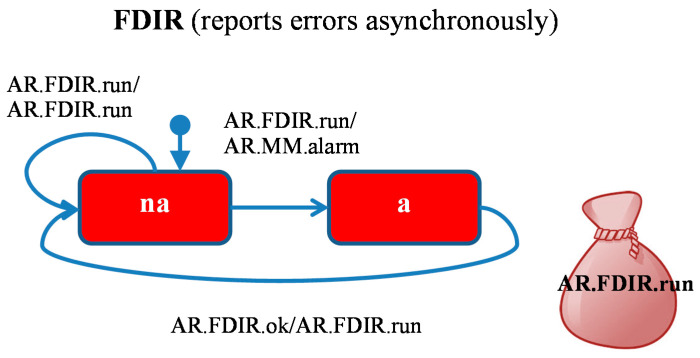
FDIR module: *AR* agent (Alarm Reporter).

**Figure 6 sensors-21-04541-f006:**
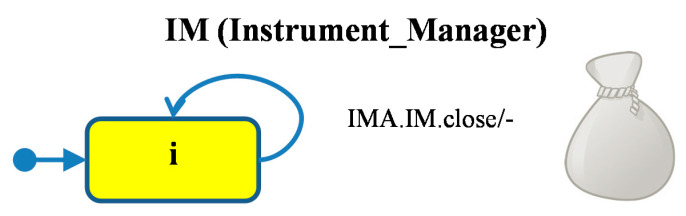
*IM* module (Instrument Manager), *IMA* (Instrument Manager Agent) starts in *MM.*

**Figure 7 sensors-21-04541-f007:**
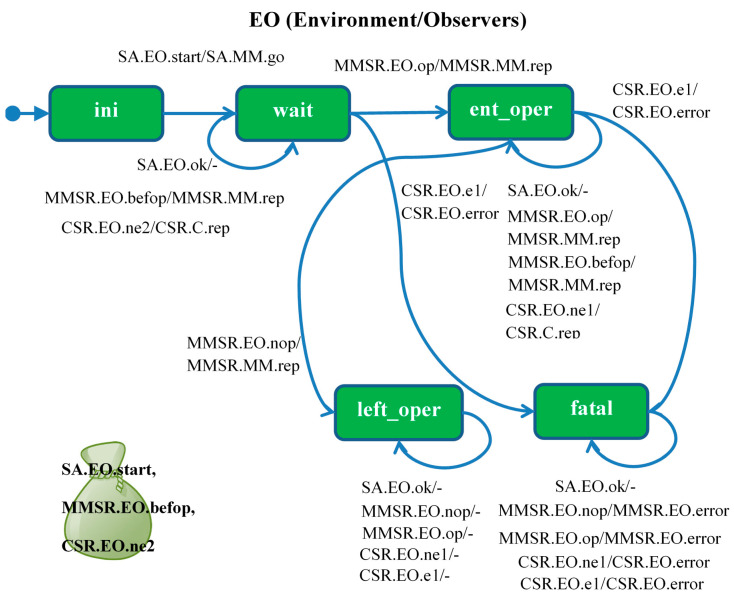
*EO* (Environment/Observers) module: agents *SA* (Staring Agent), *MMSR* (MM State Reporter), *CSR* (Counter State Reporter).

**Figure 8 sensors-21-04541-f008:**
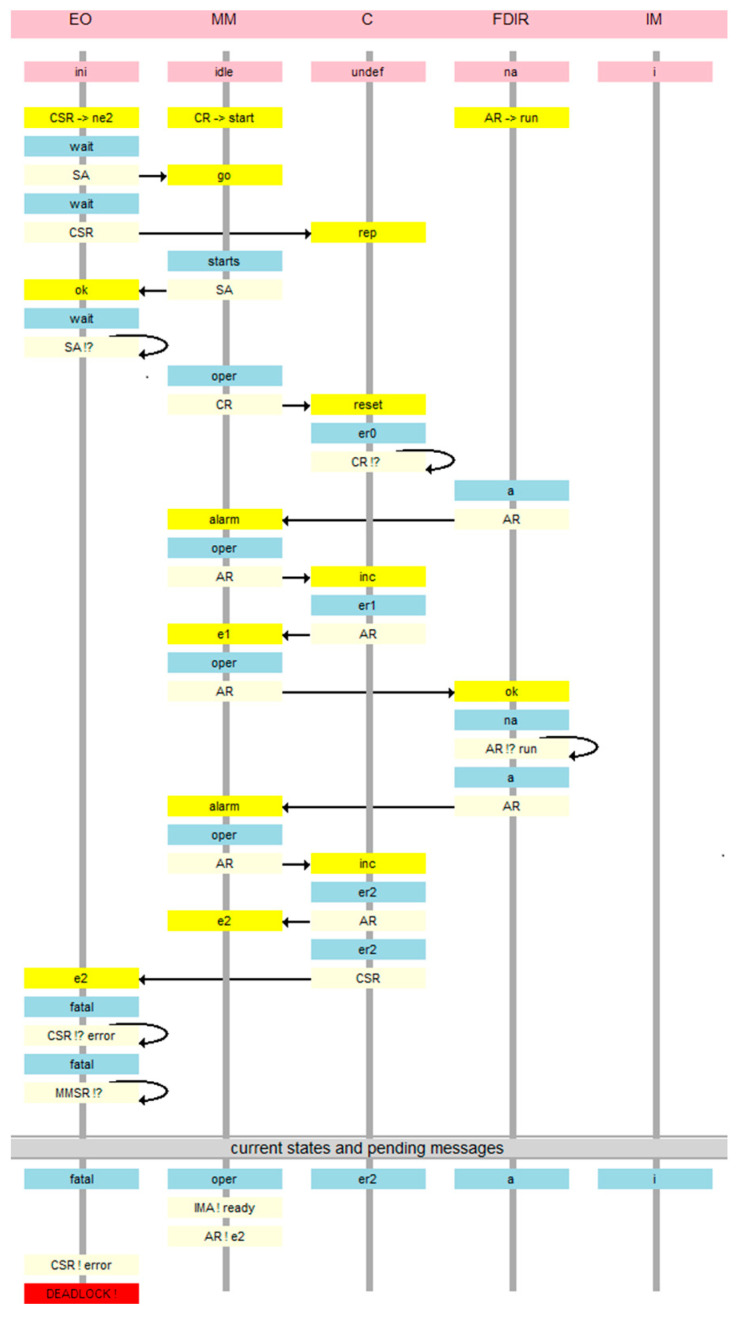
Counterexample of basic model verification in the module view.

**Figure 9 sensors-21-04541-f009:**
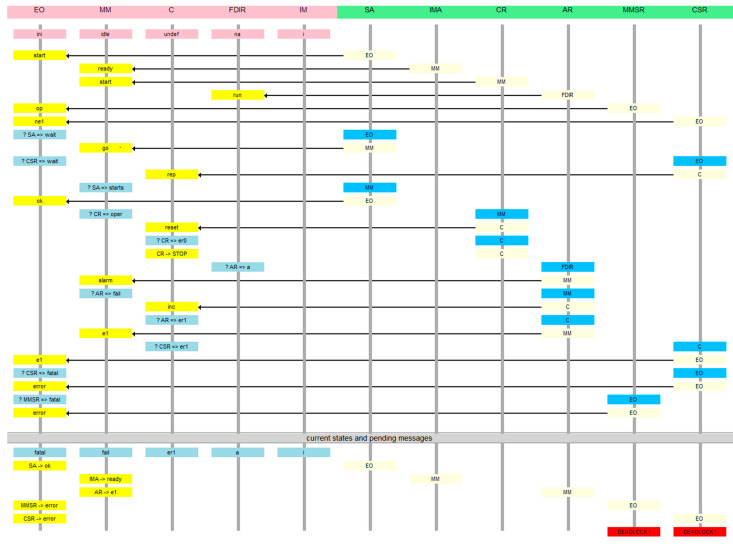
Counterexample of verification of the base model in the agent view.

**Figure 10 sensors-21-04541-f010:**
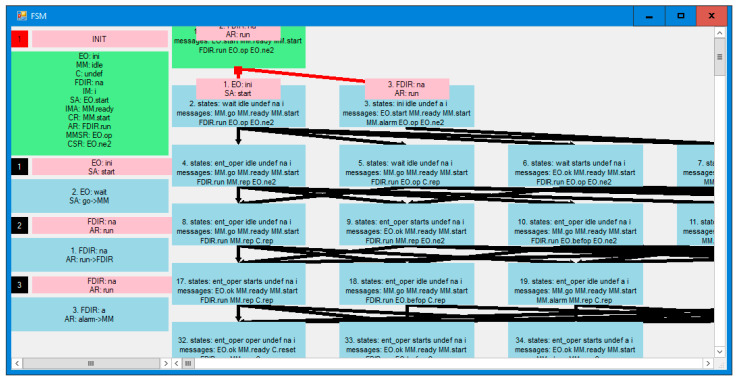
Reachability space viewer, which allows for simulation over this space.

**Figure 11 sensors-21-04541-f011:**
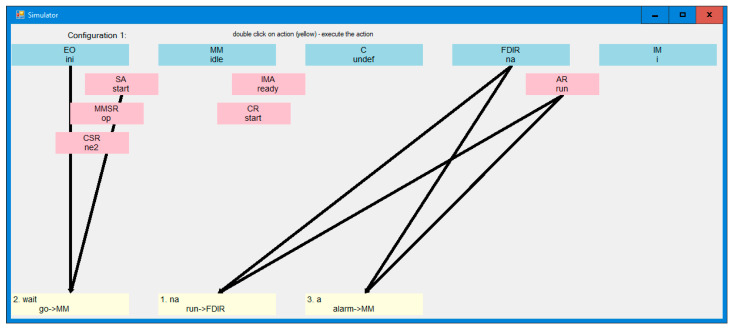
Counterexample simulation over configurations of the counterexample, or arbitrary simulation.

**Figure 12 sensors-21-04541-f012:**
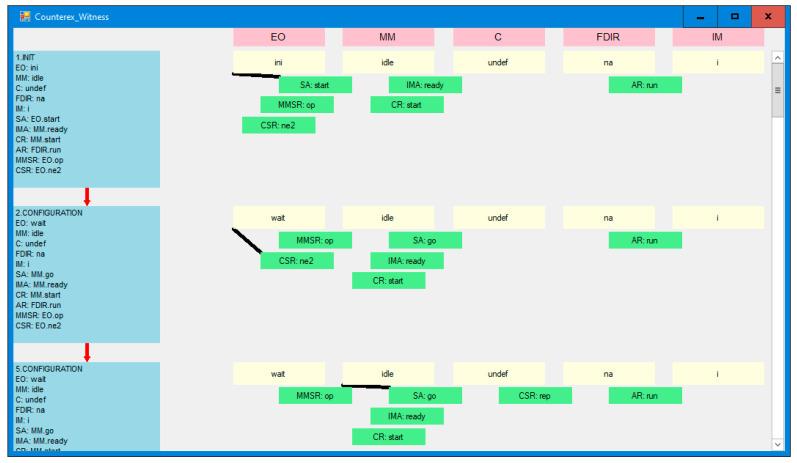
Counterexample simulator over the reachability space.

**Figure 13 sensors-21-04541-f013:**
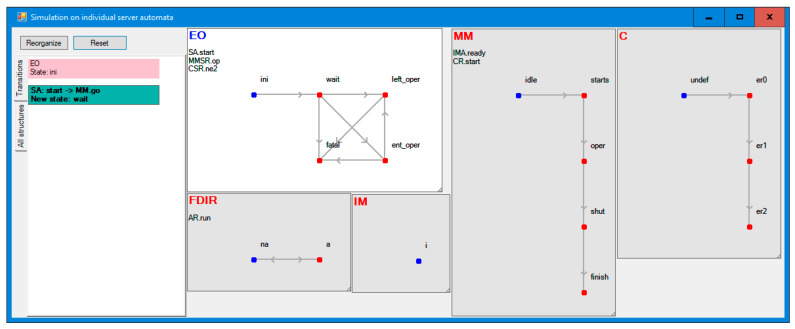
Graphic simulator over DA^3^.

**Figure 14 sensors-21-04541-f014:**
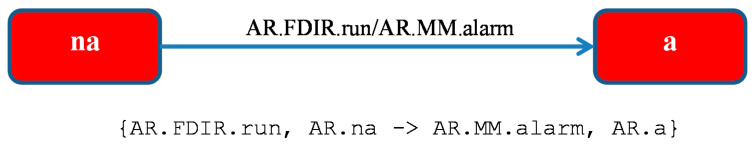
The transition in DA^3^ automaton and its textual IMDS form.

**Figure 15 sensors-21-04541-f015:**
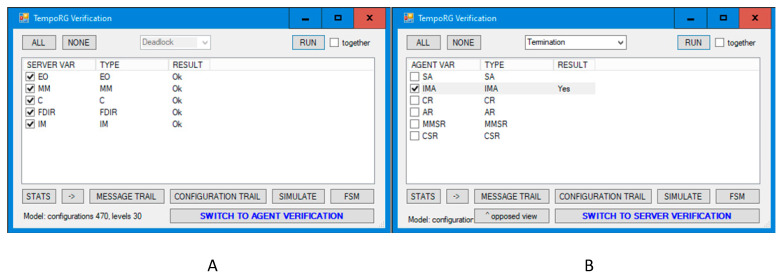
Verification of corrected version: deadlock freeness (**A**) and inevitable termination of IMA agent (**B**).

**Figure 16 sensors-21-04541-f016:**
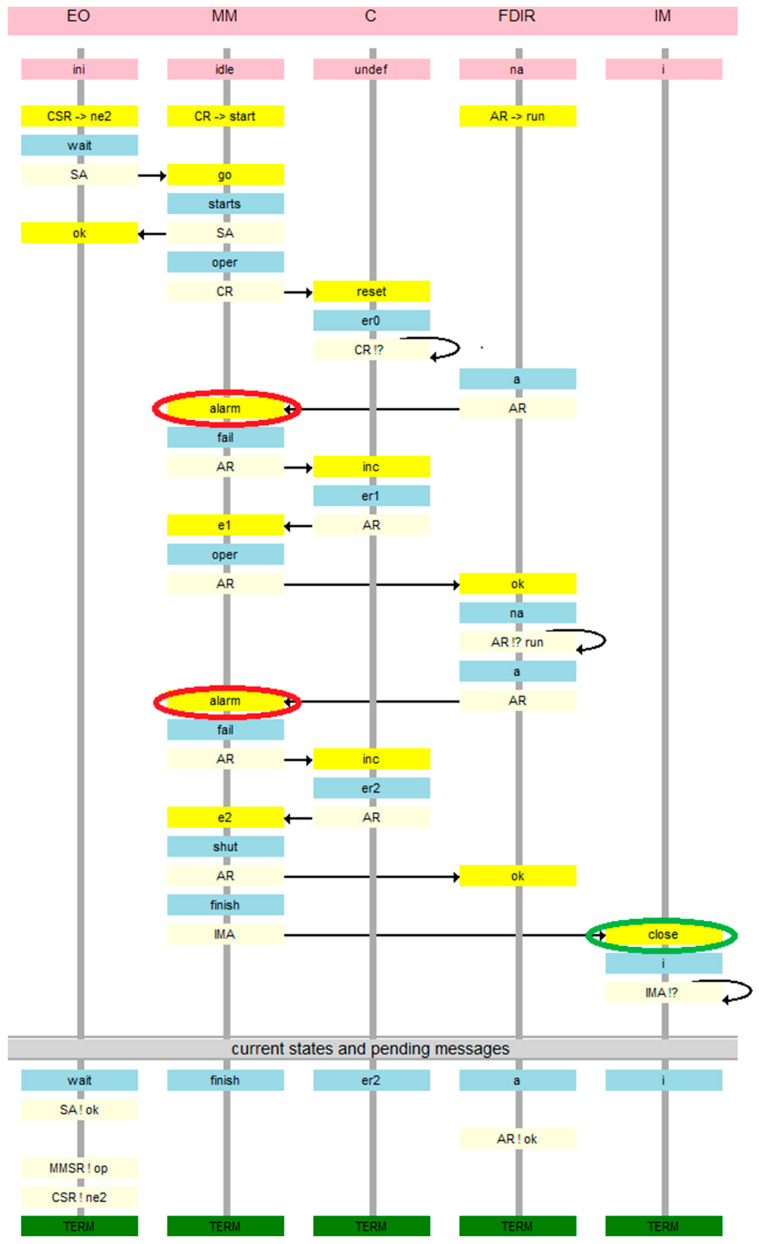
The witness of inevitable termination of *IMA* agent (node view).

**Figure 17 sensors-21-04541-f017:**
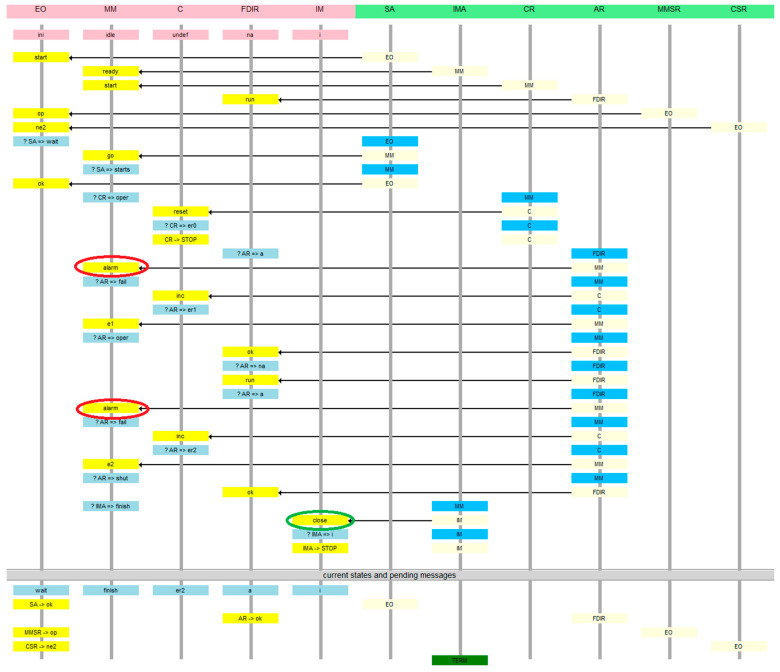
The witness of inevitable termination of *IMA* agent (agent view).

**Figure 18 sensors-21-04541-f018:**
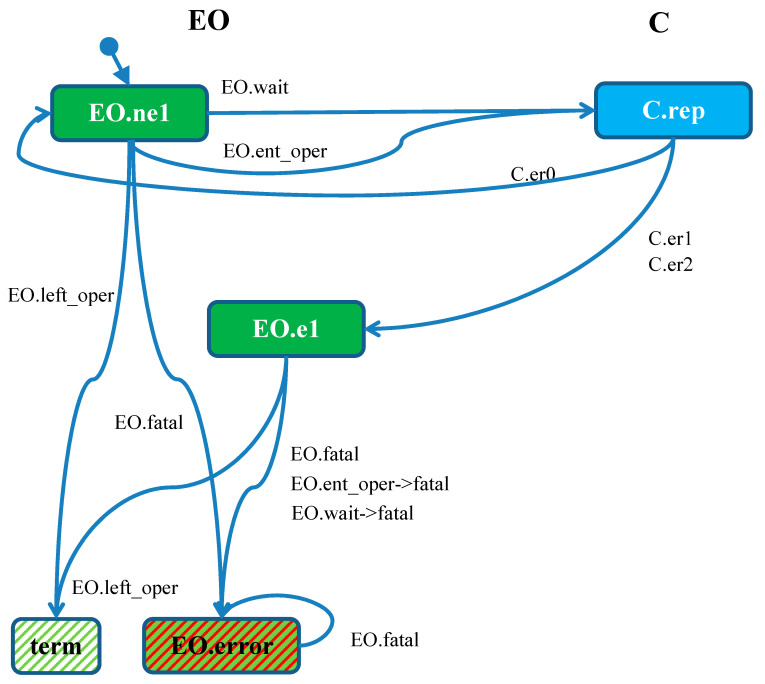
The automaton of the *CSR* observer being the base for the on-line test. In the vertices, agent name is suppressed because it is the same for all messages (*CSR*).

## Data Availability

The IMDS source code of the “naive” version, and the final version both in the node view and the agent view is supplied as supplementary files.

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
