# Peer review of "Static and Dynamic Verification of Space Systems Using Asynchronous Observer Agents"

_sensors, 2021, doi:10.3390/s21134541_

Round 1
Reviewer 1 Report
This paper presents the idea of using observers in the IMDS formalism to check the state of individual system components. The introduction states the purpose of the paper, and the relation between the paper and previous works is clearly explained. The verification can be implemented without knowing the global state of the system using the observers.
The following are my specific comments:
(1) It is stated in the Abstract that the same observers are used for formal verification and for validation of the system operation at runtime. How is the observers applied for validation of the space system operation at runtime? The authors are suggested to provide an explicit procedure for the implementation of the approach in space missions.
(2) A poor example of a system which was not formally verified is provided in the introduction. I wonder whether it is a general case. In addition, the introduction can be simplified to focus on the main contribution of the paper.
(3) It is stated in Section 2 that a verification method based on the IMDS is addressed primarily to the verification of distributed systems. It is better to list some examples for its applications on satellite control. The main advantage of the verification method should be summarized.
(4) In Section 3, an example of an observer is given, and the usual cooperation between the modules is shown as bold arrows. In my opinion, the description about the usual cooperation is ambiguous.
(5) The full names of the abbreviations SCADA, SMS, GSM and FDIR should be given right after they first appear. In addition, I cannot find Fig.5.1-Fig.5.5.
(6) In Section 6, there may be a typo error in the sentence "the verification of shows the possibility ...". The explanations of Fig.8-Fig.13 are not satisfactory. Some detailed descriptions are required if the figures are important for performance illustration.
(7) The result for the verification of the corrected model should be concluded in Section 7.
(8) The authors are suggested to summarize the main innovations of the paper explicitly in the conclusion.
Author Response
Dear reviewer,
Thank you for careful review. All the comments caused appropriate corrections, requiring changes in the text, and some of them are the reason for new fragments. The changes are clearly visible as they are prepared in registering mode. Below are responses to individual comments with references to changes in the text.
(1) It is stated in the Abstract that the same observers are used for formal verification and for validation of the system operation at runtime. How is the observers applied for validation of the space system operation at runtime? The authors are suggested to provide an explicit procedure for the implementation of the approach in space missions.
Response: The translation of the CSR automaton to the pseudocode of the agent that can be included into the running system is given and discussed in Sect. 7, page 23.
(2) A poor example of a system which was not formally verified is provided in the introduction. I wonder whether it is a general case.
Response: We do not know how general the Pathfinder case is (but the priority inversion phenomenon occurs in many projects). Designers rarely boast about mistakes, but rather keep silent about them. However, even if Pathfinder were the only space mission that was threatened with a failure as a result of a process synchronization error, the possibility of losing several billion dollars and several years of work as a result of a programming error is sufficient prerequisite for creating a methodology for formal system verification in the design phase, and later runtime verification of compliance with specification.
This text in included in the Introduction, page 1.
(2a) In addition, the introduction can be simplified to focus on the main contribution of the paper.
Response: The discussion on our research in the context of other approaches has been moved to Sect. 2.
(3) It is stated in Section 2 that a verification method based on the IMDS is addressed primarily to the verification of distributed systems. It is better to list some examples for its applications on satellite control. The main advantage of the verification method should be summarized.
Response: We have verified numerous systems, some of them are mentioned in the paper, Sect. 4. We have verified avionic systems, however we cannot give references because this project is confidential. This paper is based on our experience with verification of distributed systems, and is inspired by ESA tender materials, which is mentioned in the Introduction, page 3.
(4) In Section 3, an example of an observer is given, and the usual cooperation between the modules is shown as bold arrows. In my opinion, the description about the usual cooperation is ambiguous.
Response: The picture presents the idea of observer rather than an example (which is changed in the text).
The exchange of messages between these modules is not important from our point of view. It is important that these modules reach the correct states in the correct order. This is the case in the ESA tender, where the way of cooperation between the MM, FDIR and IM modules is not defined at all. Only alarm signals from sensors that are counted are included.
This text in included in Sect. 3, page 8.
(5) The full names of the abbreviations SCADA, SMS, GSM and FDIR should be given right after they first appear.
Response: The development of abbreviations SCADA, SMS and GSM are given, page 9 bottom and page 10 top.
The abbreviation FDIR is not elaborated anywhere in the ESA tender materials, so we treat it as a proper name.
This text is included in the footnote on page 4.
(5a) In addition, I cannot find Fig.5.1-Fig.5.5.
Response: there were wrong references to the figures in the text, corrected.
(6) In Section 6, there may be a typo error in the sentence "the verification of shows the possibility ...".
Response: thank you, corrected.
(6a) The explanations of Fig.8-Fig.13 are not satisfactory. Some detailed descriptions are required if the figures are important for performance illustration.
Response: detailed description is added for every figure, pages 15-18.
(7) The result for the verification of the corrected model should be concluded in Section 7.
Response: We check this property by simple verification for termination of IMA agent, which is expected after two alarms. This agent terminates after closing the door, which is specified as desired result in original ESA materials. The result of the verification is presented in Fig. 15 (right), and the witness sequence diagram is shown in Fig. 16 (node view) and 17 (agent view). These graphs show that, according to the ESA tender requirement, the IMA agent successfully terminates after closing the instrument door (in green ovals) in response to two alarm signals from the sensors (in red ovals).
This text is added in Sect. 7, page 19.
(8) The authors are suggested to summarize the main innovations of the paper explicitly in the conclusion.
Response: the conclusions are rebuilt accordingly, page 24.

Reviewer 2 Report
The paper showed a verification of a space system, in which signals from sensors initiate appropriate reactions. The model used highlights the actual features of commination with satellite, and between satellite modules. The crucial is asynchrony of communication and actions, and acting without any global knowledge of the state of the system. Some corrections are needed as follows.
1. The abstract is too long. Examples should be excluded from the abstract. Some sentences should be re-organised to make the abstract shorter.
2. Static verification and dynamic testing over the same model should be further explained.
3. The contribution and novelty of the work can be made more outstanding in the introduction.
4. What is the difference between a reachability graph and a network. Note that a network can also have information attached with the nodes/edge as well as dynamics.
5. More explanation about Dedan environment should be given.
6. How do you differentiate the AR.MM alarm and MMSR.MM/rep/MMSR.EO.nop in practice as well as in the model effect?
7. Figures 9, 16, 17 are blurred. Please provide figures with better quality.
8. Ref 26 should be deleted. It is not very relevant.
Author Response
Dear reviewer,
Thank you for careful review. All the comments caused appropriate corrections, requiring changes in the text, and some of them are the reason for new fragments. The changes are clearly visible as they are prepared in registering mode. Below are responses to individual comments with references to changes in the text.
- The abstract is too long. Examples should be excluded from the abstract. Some sentences should be re-organised to make the abstract shorter.
Response: the abstract is rebuilt accordingly, page 1.
- Static verification and dynamic testing over the same model should be further explained.
Response: The translation of the CSR automaton to the pseudocode of the agent that can be included into the running system is given and discussed in Sect. 7, page 23.
- The contribution and novelty of the work can be made more outstanding in the introduction.
Response: the introduction is rebuilt accordingly, page 1 and 2. The paragraphs concerning our verification approach are shifted to Sect. 2.
- What is the difference between a reachability graph and a network. Note that a network can also have information attached with the nodes/edge as well as dynamics.
Response: Model checking is performed over the reachability graph of the system, which consists of the vertices being system states and transitions which are actions. Every model checker uses a single graph for verification, on which temporal formulas are evaluated, or a similar checking mechanism is applied. Of course, in a distributed system such a graph must be elaborated as a net effect of a composition of individual components graphs. The global graph is stored in the checker, or calculated partly in non-exhaustive verification methods. However, it is never directly observed as a whole due to its size (except for tiny cases).
This text is added in Sect. 2, page 5.
- More explanation about Dedan environment should be given.
Response: The operation of Dedan is given in Sect. 6, pages 15-18.
- How do you differentiate the AR.MM alarm and MMSR.MM/rep/MMSR.EO.nop in practice as well as in the model effect?
Response: The source code of the CSR agent in IMDS is given in Sect. 7, page 21.
The automaton of CSR or MMSR agent can be programmed in a language used to implement the system. In order to conform to the formal model, the transmitted messages must be provided with the identifier of the agent in the context in which they are sent. For example, see messages in the above CSR agent process having this identifier as the first item.
This text is added in Sect. 7, page22.
- Figures 9, 16, 17 are blurred. Please provide figures with better quality.
Response: These figures are simply the output from the Dedan program, they are blurred due to pasting them into ms word. High resolution graphics will be provided separately to the final version for processing.
- Ref 26 should be deleted. It is not very relevant.
Response: It has been deleted

Round 2
Reviewer 1 Report
The quality of the manuscript is improved after the revision.
Reviewer 2 Report
I have no further comment. The paper is good now.